# A Bidirectional Versatile Buck–Boost Converter Driver for Electric Vehicle Applications

**DOI:** 10.3390/s21175712

**Published:** 2021-08-25

**Authors:** Catalina González-Castaño, Carlos Restrepo, Samir Kouro, Enric Vidal-Idiarte, Javier Calvente

**Affiliations:** 1Department of Engineering Sciences, Universidad Andres Bello, Santiago 7500971, Chile; inv.cet@unab.cl; 2Department of Electromechanics and Energy Conversion, Universidad de Talca, Curicó 3340000, Chile; crestrepo@utalca.cl; 3Electronics Engineering Department, Universidad Técnica Federico Santa María, Valparaíso 2390123, Chile; samir.kouro@usm.cl; 4Departament d’Enginyeria Electrònica, Elèctrica i Automàtica, Escola Tècnica Superior d’Enginyeria, Universitat Rovira i Virgili, 43007 Tarragona, Spain; javier.calvente@urv.cat

**Keywords:** noninverting buck–boost converter, high efficiency, wide bandwidth control, discrete-time sliding-mode current control (DSMCC), electric vehicle (EV), driver vehicle system, energy management

## Abstract

This work presents a novel dc-dc bidirectional buck–boost converter between a battery pack and the inverter to regulate the dc-bus in an electric vehicle (EV) powertrain. The converter is based on the versatile buck–boost converter, which has shown an excellent performance in different fuel cell systems operating in low-voltage and hard-switching applications. Therefore, extending this converter to higher voltage applications such as the EV is a challenging task reported in this work. A high-efficiency step-up/step-down versatile converter can improve the EV powertrain efficiency for an extended range of electric motor (EM) speeds, comprising urban and highway driving cycles while allowing the operation under motoring and regeneration (regenerative brake) conditions. DC-bus voltage regulation is implemented using a digital two-loop control strategy. The inner feedback loop is based on the discrete-time sliding-mode current control (DSMCC) strategy, and for the outer feedback loop, a proportional-integral (PI) control is employed. Both digital control loops and the necessary transition mode strategy are implemented using a digital signal controller TMS320F28377S. The theoretical analysis has been validated on a 400 V 1.6 kW prototype and tested through simulation and an EV powertrain system testing.

## 1. Introduction

Electric vehicles (EVs) are an essential part of meeting global goals on reducing the carbon footprint of vehicle emissions that contribute to climate change [1,2]. All the EVs powertrain configurations shown in Figure 1 have a common system that is formed by the battery, the power converter, and the electric motor. Each of these components has been the subject of extensive research in recent years and a high level of development to improve the performance of the automotive traction systems. However, these three components represent a tremendous research challenge given the complexity of integrating these elements in EVs application.

In EVs, the battery is generally sized by the energy requirements to allow a specific range to be reached. Still, there is not a linear relationship between car range and battery capacity because adding the weight of the battery reduces the efficiency on the road [2,3]. The battery cells for EVs are usually connected in series to meet the voltage requirements of the power converter (inverter). The connection of cells in series exponentially increases the probability of failure of the battery pack. The performance of the whole pack is limited by the weakest cell and the oversizing of the power inverter and the electric motor to ensure peak power delivery at a low state of charge (SoC) of a battery pack with a wide voltage variation at different SoC [4,5]. Thus, there is a limitation of the maximum number of battery cells that could be connected in series, and a step-up dc-dc power converter is required to reach the requirements of the inverter converter. Therefore, the power converter shown in Figure 1 is implemented using a step-up DC-DC in cascade with a DC-AC traction inverter (see Figure 2 DC-DC + DC-AC block) [6].

The induction machine (IM) and the permanent magnet synchronous machine (PMSM) are the most used [7,8,9] electric motors in EV traction applications. In the constant torque operation region (Figure 2), the maximum torque capability is determined by the current rating of the inverter [7,10]. The maximum torque at base speed (point A in Figure 2) defines the vehicle performance at starting or climbing hills [8]. The available torque at maximum speed (point B in Figure 2) in the constant power region limits the vehicle speed highways. In the last region, the torque and power reduction are due to the back-electromotive force’s increasing influence [7,10].

Connecting a DC-DC converter between the battery and the inverter allows optimizing the inverter’s DC input voltage, improving power capability, and maximizing the electric motor efficiency [11]. A bidirectional DC-DC converter can be used to control the voltage at the input of the inverter according to the motor speed. In this way, the converter can optimize the efficiency of the inverter (modulation index MI=1 achieves it) in a wider range of operating speeds, as can be seen in Figure 2 (see DC-DC + DC-AC block) [12,13].

In [12], the authors used an interleaving half-bridge bidirectional converter to regulate a variable DC-bus voltage, showing the efficiency improvement both in the step-up converter and inverter. A detailed inverter loss model is developed in [13], where a variable DC-bus voltage closely related to the rotational motor speed significantly improves the inverter efficiency for voltages above the battery voltage. Despite proposals of using composite topologies for high step-up gain [14] or flying capacitors topologies to reduce inductor size [15,16], the most commonly used converter for this application has been the bidirectional half-bridge [12,13,17,18,19,20,21,22,23,24] and boost [25] converters. In [18], the authors proposed a three-level version of this converter to use lower breakdown voltage MOSFETs. A coupled inductor in each phase is proposed to increase the power [19].

An interleaved zero voltage switching (ZVS) version, included in multifunctional power electronic interface and operating at 60 kHz, is presented in [20], achieving high-efficiency measurements. Integration of this bidirectional converter in a new topology is proposed in [24] for a hybrid electric vehicle system. This converter interfaces between two different voltage values corresponding to the battery system and a DC-bus. It is worth noting that this bidirectional converter operates as a boost- or buck-converter depending on whether the motor is in driving or regenerative mode [26]. Therefore, this electric drive topology is more suitable for highway driving cycles (see Figure 2), reducing the system’s efficiency under an urban driving cycle. The latter is because the inverter efficiency cannot be guaranteed under low speeds since the boost converter cannot reduce the DC-bus voltage below the battery voltage [6,27]. A converter with step-up and step-down characteristics, not only will extend the efficient range to urban driving cycle, but will also add more flexibility in designing both battery and inverter.

The noninverting buck–boost converter with coupled inductors known as the versatile buck–boost converter and shown in Figure 3a, could be an excellent candidate to optimize the global efficiency of the system. It has many advantages, such as noninverting voltage step-up and step-down characteristic in both motor operating modes, high efficiency, wide bandwidth [28], and input or output currents regulation because of their low ripple values [29,30]. It provides smooth transitions between buck and boost operating modes due an hysteresis PWM control strategy used to activate the controlled switches [31]. In addition, the introduction of an RC damping network in parallel with the intermediate capacitor, combined with the coupled-inductors, eliminates the right half-plane zero that limits the closed-loop bandwidth of the step-up converters [28]. All the advantages mentioned above have allowed its use in different fuel cell hybrid power systems [32,33,34] and deepen on various digital current control techniques [35,36,37].

A novel bidirectional version of the versatile buck–boost converter is presented to extend its use in electric vehicle applications. This new converter shares some similarities with their previous ones (see Figure 4) to preserve all the advantages of the versatile converter. However, two significant changes to match the hard-switching high-voltage bidirectional EV application are included in this new version. The first one corresponds the use of the Silicon Carbide (SiC) devices that extend the operation at high-voltage with low switching losses [23,38,39,40]. The second one corresponds with a redesign of the coupled inductors to reduce the parasitic winding-to-winding capacitance [41]. Other important aspects that differentiate the converter presented in this work from the existing ones are summarized in Figure 4.

This work presents a novel high-voltage bidirectional buck–boost converter with digital control that allows the regulation of the high-voltage DC-bus for EV applications. This voltage regulation and the energy flow between the battery and the motor drive are managed by means of a two-loop digital current control strategy, which facilitates the hysteresis transition between voltage step-up and step-down modes. The resulting control provides output voltage regulation in the presence of variations in output voltage and load power. The whole system is tested experimentally in a 1.6 kW prototype applied to a resistive load and an EV hardware emulation platform. Based on Figure 4 and the state of the art, the main contributions of this paper can be summarized as follows:A novel high-voltage high-switching bidirectional converter is presented. This new converter has step-up and step-down characteristics in both current directions to extend the EV traction inverter efficiency under a wide range of speeds. This converter guarantees a high power conversion efficiency for EV powertrain applications due to silicon carbide (SiC) devices and the design with a low winding-to-winding parasitic capacitance of the coupled inductor. It can operate in boost or buck mode.A two-loop digital control design with a current (inner loop) controller and a voltage (outer loop) controller regulate the DC-bus voltage during traction and regenerative modes. The proposed controller ensures zero steady-state voltage error and fast transient responses to the voltage reference and power variations.A DSMCC control is proposed for the inner loop of the voltage feedback outer loop. The proposed controller ensures fast-tracking of the control set-points and low steady-state error under demanding tests that include system start-up and dc bus voltage reference with small and large variations. It is the first time that the DSMCC control strategy is used for the versatile buck–boost converter.

This paper is organized as follows: Section 2 presents an analysis of the coupled inductors buck–boost converter with the goal of obtaining the inductor current slope equations. Section 3 describes the DSMCC technique implemented for the inner control loop. This section also includes the outer voltage feedback loop analysis, which is based on a PI controller. Simulations and experimental results of the current control technique under startup, small and large variations, and using an EV emulator are presented and discussed in Section 4. Finally, the main conclusions and the remaining challenges for the future are summarized in Section 5.

## 2. Bidirectional Noninverting Coupled-Inductor Buck–Boost Converter

The converter scheme depicted in Figure 3a is composed of two half-bridge MOSFETs, an $R_{d}C_{d}$ damping network connected in parallel with the intermediate capacitor *C*, a constant input voltage $V_{g}$, and a resistance load $R_{o}$. In addition, the coupled inductor has a unitary ideal turns ratio $N_{2}/N_{1}$, a coupled coefficient k=0.5, a mutual inductance *M* and equal values for the primary ($L_{1}$) and secondary ($L_{2}$) self-inductances ($L=L_{1}=L_{2}$). In the analysis, a continuous conduction mode (CCM) operation is considered, with no parasitic effects and a switching frequency much higher than the converter’s natural frequencies. The use of the state-space averaging (SSA) method to model the converter leads to the following set of differential equations [37]:(1)$$\frac{di_{g}\left(t\right)}{dt}=\frac{L\left(V_{g}-v_{c}\left(1-u_{1L}\right)\right)-M\left(v_{o}-v_{c}u_{2H}\right)}{L^{2}-M^{2}}$$
(2)$$\frac{di_{L}\left(t\right)}{dt}=\frac{M\left(V_{g}-v_{c}\left(1-u_{1L}\right)\right)-L\left(v_{o}-v_{c}u_{2H}\right)}{L^{2}-M^{2}}$$
(3)$$\frac{dv_{c}\left(t\right)}{dt}=\frac{1}{C}\left(-i_{L}u_{2H}+i_{g}\left(-u_{1L}+1\right)-\frac{1}{R_{d}}\left(v_{c}-v_{cd}\right)\right)$$
(4)$$\frac{dv_{cd}\left(t\right)}{dt}=\frac{v_{c}-v_{cd}}{C_{d}R_{d}}$$
(5)$$\frac{dv_{o}\left(t\right)}{dt}=\frac{i_{L}}{C_{o}}-\frac{v_{o}}{R_{o}C_{o}}$$

In the scheme of Figure 3, the duty cycle $d_{1}\left(t\right)$ is used to activate the switch $Q_{1}$ and $Q_{2}$ for boost mode. $Q_{3}$ and $Q_{4}$ are switched with the duty cycle $d_{2}\left(t\right)$ for buck mode. The activation signals $u_{1H}$ and $u_{1L}$ are for the half-bridge composed of $Q_{1}$ and $Q_{2}$, and the activation signals $u_{2H}$ and $u_{2L}$ are for the half-bridge composed by $Q_{3}$ and $Q_{4}$. $u_{1H}$ and $u_{1L}$ operate in a complementary manner while $u_{2H}$ is set at 1 and $u_{2L}$ is set at 0, in boost mode. Otherwise, $u_{2H}$ and $u_{2L}$ operate in a complementary manner while $u_{1H}$ is set at 1 and $u_{1L}$ is set at 0, in buck mode. The duty cycles are computed considering a variable control u(t), where $u\left(t\right)\;=\;1+d_{1}\left(t\right)$ in boost mode and $u\left(t\right)\;=\;d_{2}\left(t\right)$ for buck mode [28]. Figure 3b shows the hysteresis transition method avoids oscillations in the transitions between buck, boost, and buck–boost working modes [31]. The aim of this analysis is to find the converter’s current output slope $\frac{di_{L}}{dt}$ in each operation mode (buck or boost) to design the digital inner current programmed controller. The output current has a periodic triangular waveform where the current rises with a slope of $m_{1}$ and falls with a slope $-m_{2}$. Table 1 presents the converter output current waveform slopes based on Equation (Equation 2) for the boost and buck modes.

## 3. Digital Control for Output Voltage Regulation

The control method implemented to regulate the converter’s output voltage is a two-loop digital control. This strategy allows smooth transitions between motoring and regenerative braking operations and during the DC-bus voltage reference changes. The digital control has the advantage of simplifying the implementation of complex control strategies, the soft start of the converter, higher robustness to noise, and flexibility in design without the need to make any component or hardware changes [42]. In addition, it allows the integration of the hysteresis mode transition strategy in the digital controller, making the implementation and tuning of this transition strategy easier. The digital control proposed has an inner current programmed controller with an outer voltage feedback loop (PI compensator).

The current control loop must present a fast dynamic response to reduce the transient response between buck and boost modes. This can be achieved using discrete-time sliding-mode current control (DSMCC) for the output current $i_{L}$, taking into account the converter dynamics.

### 3.1. Discrete-Time Sliding-Mode Current Control

This work presents a fixed switching DSMCC control for the bidirectional noninverting buck–boost converter. This discrete sliding control has been presented for a boost converter in [43] and a buck converter [44]. In this control strategy, the DSMCC aims to compute the variable control u[n] in the *n*-th time sample period that ensures the control surface (Equation Equation 6)) is reached in the next sampling period ($f_{samp}=f_{s}$).
(6)$$s\left[n\right]=i_{Lref}\left[n-1\right]-i_{L}\left[n\right].$$

The Euler approximation leads to the following discrete-time output current expression, assuming the averaged model that the converter’s current output slope $\frac{di_{L}}{dt}\approx \frac{i_{L}\left[n+1\right]-i_{L}\left[n\right]}{T}$
(7)$$i_{L}\left[n+1\right]=i_{L}\left[n\right]+T\left(m_{1}+m_{2}\right)d_{x}\left[n\right]-m_{2}T.$$

Hence, the resulting expression of the duty cycle is
(8)$$d_{x}\left[n\right]=\frac{1}{(m_{1}+m_{2})T}\left[i_{Lref}\left[n\right]-i_{L}\left[n\right]\right]+\frac{m_{2}}{m_{1}+m_{2}}$$
where *x* in Equation (Equation 8) corresponds to the operating mode of the bidirectional buck–boost converter (*x* = 1 for boost mode, *x* = 2 for buck mode), and $i_{Lref}\left[n\right]=i_{L}\left[n+1\right]$, using the expressions for $m_{1}$ and $-m_{2}$ for the output current slopes from Table 1 in Equation (Equation 8). The expression $m_{1}+m_{2}$ is obtained from Table 1 for each converter operation mode, yielding
(9)$$m_{1}+m_{2}=\left\{\begin{array}{ccc} \frac{MV_{c}}{L^{2}-M^{2}} & \; & \mathrm{for}\mathrm{boost}\mathrm{mode}\\ \frac{LV_{c}}{L^{2}-M^{2}} & \; & \mathrm{for}\mathrm{buck}\mathrm{mode}. \end{array}\right.$$

For $m_{2}/\left(m_{1}+m_{2}\right)$, it is given by
(10)$$\frac{m_{2}}{m_{1}+m_{2}}=\left\{\begin{array}{ccc} \frac{-M\left(V_{g}-V_{c}\right)+L\left(V_{o}-V_{c}\right)}{MV_{c}} & \; & \mathrm{for}\mathrm{boost}\mathrm{mode}\\ \frac{-M\left(V_{g}-V_{c}\right)+LV_{o}}{LV_{c}} & \; & \mathrm{for}\mathrm{buck}\mathrm{mode}. \end{array}\right.$$

In this control method, the output current $i_{L}\left(t\right)$ and voltages are sampled at the beginning of each switching period, then, at the end of the switching cycle, $i_{L}\left[n\right]=i_{Lref}\left[n-1\right]$. The steady-state duty cycle from the equivalent control law (Equation Equation 8)) can be obtained by substituting the voltage of the intermediate capacitor $v_{c}$ by $V_{g}$ for buck mode and by $v_{c}=V_{o}$ for boost mode in Equation (Equation 10). In steady-state, the duty cycle is $U=m_{2}/\left(m_{1}+m_{2}\right)$, thus, the variable control u[n] can be written as
(11)$$u\left[n\right]=\frac{1}{(m_{1}+m_{2})T}\left(i_{Lref}\left[n\right]-i_{L}\left[n\right]\right)+U_{n}$$
where $U_{n}=U$ for buck mode and $U_{n}=1+U$ for boost mode. The schematic diagram of the DSMCC is depicted in Figure 5.

### 3.2. Digital Proportional-Integral Voltage Control

In order to guarantee dc bus voltage regulation, it is necessary to add a slower outer voltage control loop. With this new loop, the switching converter can be operated as a controlled current source due to the control DSMCC ensures the load current will follow the current reference. Therefore, the current-controlled buck–boost converter is operated as a current source that allows driving the energy consumption of the load. From the point of view of the dc voltage control loop, voltage variations with power constant should be compensated charging or discharging the dc bus capacitor [45]. Hence, this PI control is designed to consider the filter output capacitor value $C_{o}$ or the capacitor in the dc-link for an EV powertrain $C_{bus}$. The transfer function of the PI voltage controller in the Laplace domain can be written as follows
(12)$$G_{vpi}\left(s\right)=K_{pv}+\frac{K_{iv}}{s}.$$

The output current reference to output voltage transfer function is obtained from Equation (Equation 5)
(13)$$H_{voiL}\left(s\right)=\frac{v_{o}\left(s\right)}{i_{L}\left(s\right)}=\frac{R_{o}}{R_{o}C_{o}s+1}$$

The loop-gain of the external closed loop voltage can be written as:(14)$$G\left(s\right)=H_{voiL}\left(s\right)G_{vpi}\left(s\right)H_{v}\left(s\right)e^{-sT_{m}},$$
where $H_{v}\left(s\right)$ represents the sensor gain. The term $e^{-sT_{m}}$ represents half switching cycle delay, $T_{m}=T/2$. Then, the controller transfer function (Equation (Equation 12)) can be expressed in the z domain using the forward Euler method, as follows
(15)$$G_{vpi}\left(z\right)=K_{pv}+\frac{K_{iv}T_{samp}}{z-1}.$$
where $T_{samp}=1/f_{samp}$. The forward Euler method is used to find the recurrence equation of the discrete-time integral PI control
(16)$$\begin{array}{cc} i_{Lp}\left[n\right] & =K_{pv}e_{v}\left[n\right]\\ i_{Li}\left[n\right] & =K_{iv}T_{samp}e_{v}\left[n\right]+i_{Li}\left[n-1\right]\\ i_{Lref}\left[n\right] & =i_{Lp}\left[n\right]+i_{Li}\left[n\right].\; \end{array}$$
where
(17)$$\begin{array}{cc} K_{pv} & =C_{o}2\pi f_{c} \end{array}$$
(18)$$\begin{array}{cc} K_{iv} & =\frac{K_{pv}}{T_{i}} \end{array}$$
(19)$$\begin{array}{cc} T_{i} & \geq \frac{10}{2\pi f_{c}} \end{array}$$
can be obtained from Equations (Equation 12) and (Equation 14), taking into account that the zero of Equation (Equation 12) is placed lower than one decade below $f_{c}$, which represents the crossover frequency (CF). The value of the crossover frequency for the voltage loop ($f_{c}$) should be lower than that of the current loop. Hence, a $f_{c}=2500$ Hz was selected for the voltage feedback loop.

Figure 6 depicts the Bode plots of simulated (PSIM) and experimental voltage loop gain under different operation modes (boost and buck) for the versatile buck–boost converter with a gain of the measurement system $H_{v}\left(s\right)=0.044$. These Bode plots show a similar behavior at a low frequency of the magnitude plot for the experiment and the simulation. For a quantitative evaluation, the CF and the phase margin (PM) are listed in Table 2. From this table, it can be concluded that the closed-loop system is stable.

## 4. Simulation and Experimental Results

Validation of the proposed current control strategy is performed on a 1.6 kW versatile buck–boost converter. A Texas Instruments TMS320F28377S Digital Signal Processor (DSP) is used to implement the proposed control algorithm to calculate the variable control *u* and the hysteresis buck–boost transition method introduced in [31], which was employed to compute the duty cycle values. These duty cycles allow the PWM generation using a symmetric triangular signal to get the activation signals of the MOSFETs switches. In addition, a direct voltage source AMREL SPS800X13-K02D is used as a power supply for the input voltage of the buck–boost converter shown in Figure 3 and whose parameters are listed in Table 3. The design guides of the versatile converter are described in detail in [41].

### 4.1. System Startup

The simulated and experimental results for the system startup in closed-loop are given in Figure 7. The voltage reference $V_{oref}$ is increased from 0 V in each switching period during 12 ms until 293 V with an input voltage ($V_{g}$) of 200 V and 350 V for boost and buck mode, respectively. It must also be noted that during the startup in boost mode, the system begins in buck mode and ends in boost mode in steady-state. Therefore, this experiment exhibits a smooth transition between the buck and boost operating modes. The experimental results demonstrate that the voltage output is well regulated in all the operation modes. In addition, a good agreement can be observed between the experimental measurement and the simulated with a fast and soft startup.

### 4.2. Small-Signal Response to Output Voltage Reference Variation

Figure 8 and Figure 9 show the small-signal control loop response to small output voltage reference changes during the boost and buck operation, respectively. The input voltage is set at 200 V with a constant resistive load $R_{o}=$ 200 Ω for all the study cases. In boost mode (Figure 8), the output voltage reference changed between 294 V and 296 V. While in buck mode, the output voltage reference changes between the values of 98 V and 100 V, as shown in Figure 9. The dc component in Figure 8 and Figure 9 have been removed to comprise the ±2 V step change in the output voltage reference. These results show that the output voltage is well regulated to its desired reference, and the output and input current are increased or decreased when the voltage reference changes to recover the converter’s operating point. The figures also demonstrate a good agreement between the experimental and simulation results with a short outer voltage transient of around 400 μs, which validates the proposed control method’s satisfactory operation.

### 4.3. Large-Signal Response to Output Voltage Reference Variation

Figure 10 and Figure 11 compare the large-signal response when the output voltage has a ±20 V step change. The figures show simulation and experimental waveforms of the input and output current and output voltage. Figure 10 depicts the response when the converter operates in boost mode. The output voltage reference has been changed between 293 V and 313 V. The dc component in the experimental and simulation results has been removed to appreciate the output voltage variation in boost mode. The results for buck mode are shown in Figure 11, where the output voltage reference has been changed between 100 V and 120 V. From these figures, for both control methods, the transient average current output value was successfully limited to ±4 A, which is the rated output current of the converter. These results confirm the direct relationship between the output voltage response time with the output filter capacitor value. It should be remembered that the slew-rate (SR) is defined in this case as SR=i/$C_{o}$ [V/μs], where *i* is the instantaneous current through the capacitor $C_{o}$. Therefore, the response time to step output variation depends on the SR parameter. Note that the measured current $i_{L}$ follows the current reference accurately. Some differences are presented between the simulated and experimental results concerning the input current. These differences are because the converter does not control this current and its dynamic depends on the dc power supply internal control. Again, a good agreement between the experimental and simulation results is observed.

### 4.4. Experiments with an EV Powertrain System Emulation

Energy management system in auxiliary supply in EV topologies and the use of dc-dc converter as an interface between the primary energy source and high-voltage powertrain are some of the applications for EVs. This application can be studied using a powertrain emulation system or simulation [46]. Some experiments are carried out considering the experimental PMSM platform described in Figure 12 that emulates an EV powertrain. This system is composed of two permanent-magnet synchronous motors (PMSMs). One of them (LSRPM 100 L) works as a traction motor with a maximum power of 4.5 kW, and the other one (LSRPM 90 SL) works as a controlled torque load with a maximum power of 3 kW. In order to verify the correct operation of the whole system shown in Figure 12, a test with a third of the total power is tested in this work. To increase the system’s operating power, it will be necessary to connect two more converters in parallel which is possible since they are current controlled modules. The traction part is controlled by a universal variable speed ac drive (SP2202), and it is fed using the buck–boost converter described in this work. The battery is emulated using a DC power supply (AMREL SPS800X13-K02D) connected in parallel with an electronic load to absorb the current in the case of regenerative mode. This converter is connected between the battery emulator and the output filter capacitor $C_{o}$ (R75PW44704030J). Subsequently, the traction motor is mechanically coupled to the motor that emulates the load (EV behavior). This traction motor is controlled according to a speed profile provided by a specific driving cycle. On the other hand, the load motor is controlled by a universal variable speed ac drive (SP1405) to follow a torque reference based on the vehicle dynamics. An EV powertrain system model has been implemented in the PSIM software with the parameters of the PMSMs listed in Table 4. To startup, the system with an initial voltage to fed the unidrive SP2202, a soft-starting of the dc-dc converter is implemented by the algorithm as it was previously described, and the simulated and experimental results are shown in Figure 13. During the startup, the reference voltage $v_{oref}$ changed from 0 V to the final desired output voltage value with a short transient around 0.9 s. The switching frequency for the inverter (SP2202) is 16 kHz. Figure 13a,b shows the startup response in boost mode with $V_{g}$ =200 V and steady-state the output voltage value of 350 V. In this experiment, the currents have an average value of 0 A in steady-state because the motors are not operating during the startup; however, the inverter remains switched because it has a 200 V power supply terminals. Figure 13c,d shows the steady-state converter response in buck mode with an input voltage ($V_{g}$) of 400 V and output voltage ($v_{o}$) of 300 V. The figures also show good agreement between the experimental data and the simulation results.

Figure 14 shows the transient response with a 450 W step change in the load power, setting the speed reference in 500 rpm and the torque value in 3.77 Nm to obtain a dc bus power demand of 300 W. Later, the speed reference changes to 1250 rpm to obtain a dc bus power demand of 750 W. As a result, the output current changes from 1 A to 2.5 A gradually while the output voltage is regulated at 300 V. In the experimental and simulated results of Figure 15, the converter can be seen working in boost mode with $V_{g}$ =200 V and bidirectional power flow. The speed of the traction motor is set to 500 rpm, and the torque of the load motor to 7.53 Nm to get a dc bus demand of 600 W. The dc component (300 V) was removed to appreciate the ±20 V step change in the output voltage reference. Figure 15a,b shows the results when the output voltage is changed from 300 V to 320 V. Consequently, the current output $i_{L}$ quickly goes to 4 A. Figure 15c,d shows the results when the output voltage is changed from 320 V to 300 V with a step change, the current output $i_{L}$ decreases to −3 A. This current is limited above the rated current (−4 A) due to the limitation of the source is 13 A. The battery is simulated with a dc source (AMREL SPS800X13-K02D ) in parallel with an electronic load (EA-ELR 9750-44 3U) (see Figure 12) in resistance mode to absorb 6 A, and it can absorb the current when the output voltage is decreased. The output voltage has a 20 V step change over 28 ms and has a −20 V step change at the output voltage over 12 ms. This time is different for each case because the unidrive SP2202 has an input filter capacitor of 2870 μF. Accordingly, this time depends on the filter capacitor value and the instantaneous current through the capacitor during the charging or discharging due to the output voltage step changes. Finally, there is a qualitatively good agreement between the simulated and the experimental results in the EV powertrain system emulation for the proposed controller and all the converter operation modes.

## 5. Conclusions

This paper proposed the bidirectional versatile buck–boost converter modified to operate at high voltage. This converter is an alternative to conventional topologies based on the boost converter in electric vehicle applications. The versatile converter has been located between the battery and traction inverter to regulate the dc bus in electric vehicle powertrains. The use of a high-efficiency step-up/step-down converter can improve the performance efficiency of the EV powertrain. This improvement includes an extensive range of electric motor speeds, which comprises urban and highway driving cycles. The proposed dc-dc bidirectional buck–boost converter is responsible for the dc bus voltage regulation through an outer voltage feedback loop and an inner current programmed controller. A Texas Instruments TMS320F28377S DSC is used to implement the digital control loops. The digital implementation of this current controller has allowed to include a dead-zone avoidance technique that effectively has suppressed very effectively undesirable nonlinear phenomena in the buck–boost mode transitions such as sub-harmonics or other undesirable nonlinear phenomena. The theoretical analyses have been validated using simulations and experimental tests performed on a 400-V 1.6-kW prototype. The current controller allows regulating the traction dc bus during motoring and regenerative brake conditions. The system presents zero steady error and fast transient response in the start-up for dc bus voltage reference changes and under realistic conditions using an EV powertrain system emulation. The experimental results are in good agreement with the simulation and the theoretical predictions. Future works will address the parallelization of power converters to increase the operating power of the system.

## Figures and Tables

**Figure 1 sensors-21-05712-f001:**
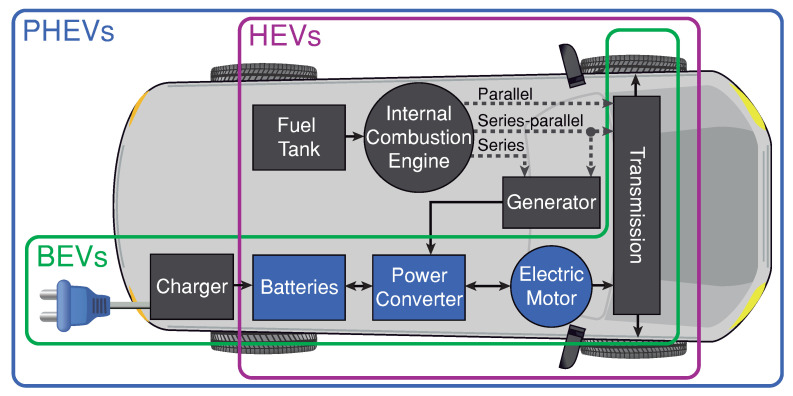
EVs powertrain configurations: hybrid electric vehicles (HEVs), plug-in hybrid electric vehicles (PHEVs) and battery electric vehicles (BEVs).

**Figure 2 sensors-21-05712-f002:**
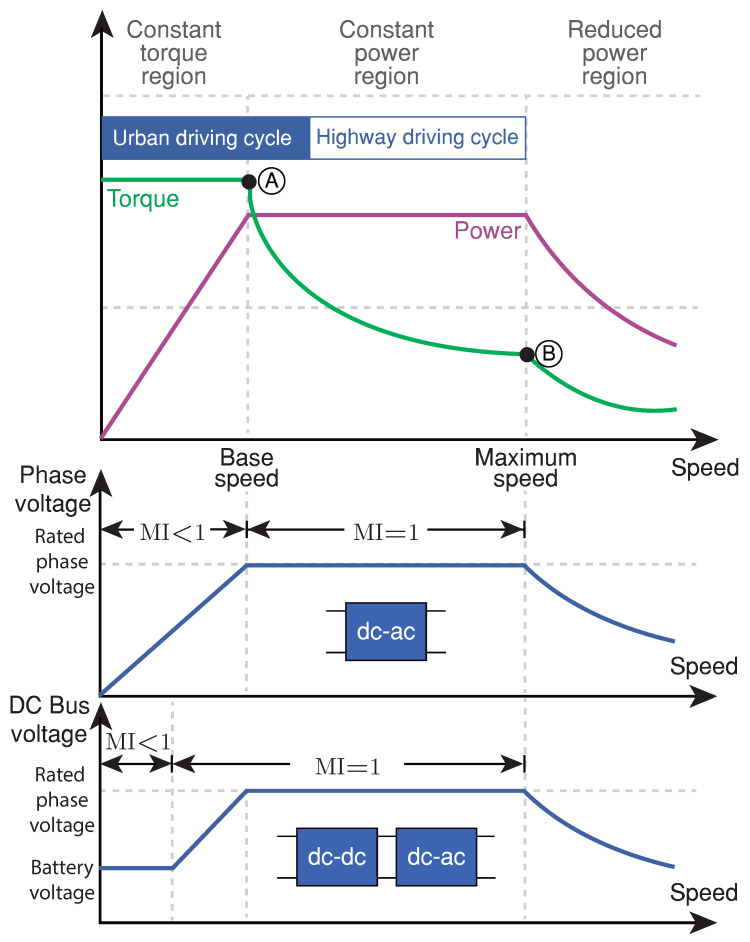
Torque and power requirements for the EV drive systems.

**Figure 3 sensors-21-05712-f003:**
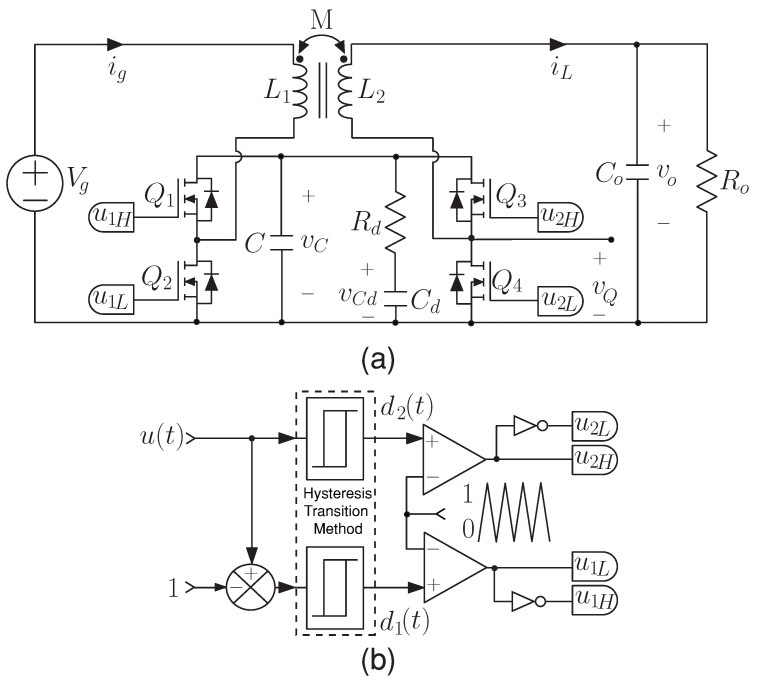
Schemes of (**a**) the buck–boost converter, (**b**) switch signals generation.

**Figure 4 sensors-21-05712-f004:**
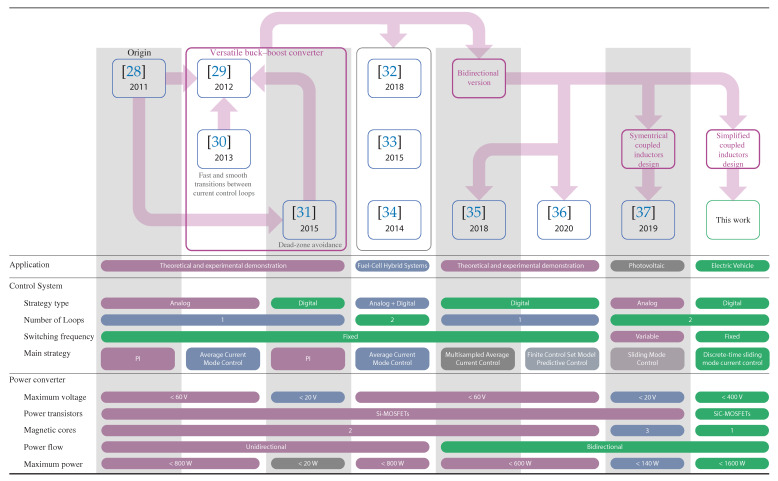
Versatile buck–boost converter evolution throughout all the investigations carried out to date.

**Figure 5 sensors-21-05712-f005:**
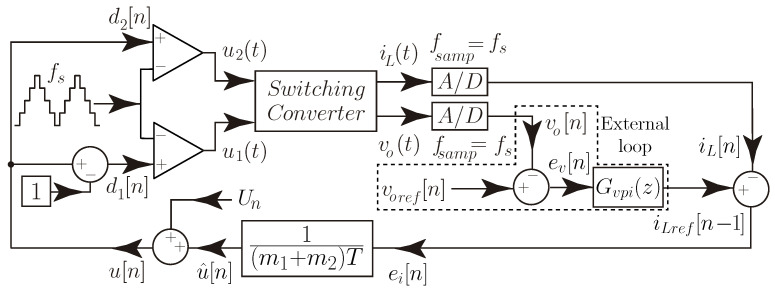
Schematic diagram of the two-loop control using DSMCC method.

**Figure 6 sensors-21-05712-f006:**
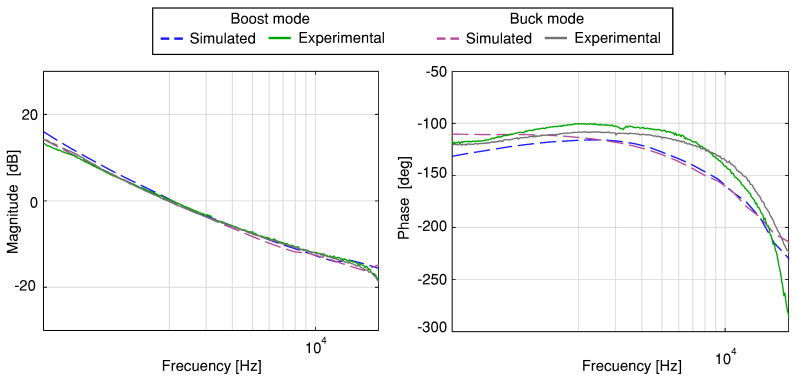
Simulated and experimental voltage loop gain Bode plots of the buck–boost converter.

**Figure 7 sensors-21-05712-f007:**
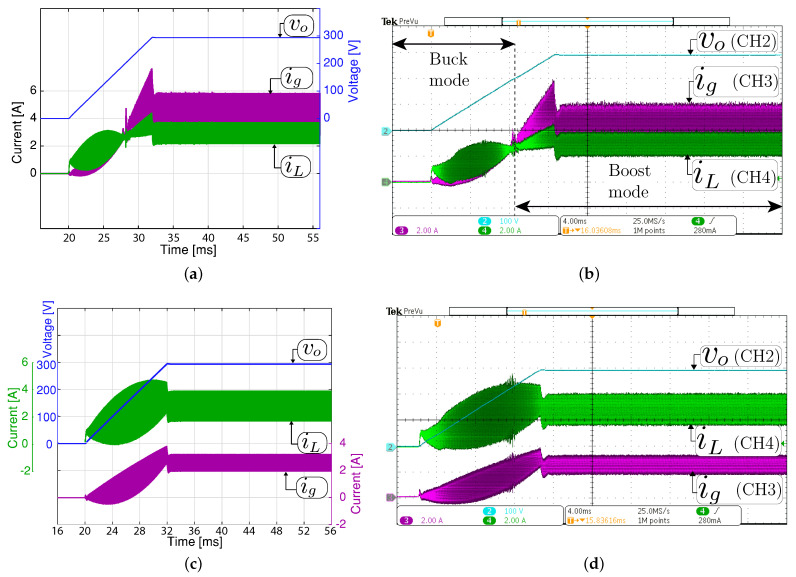
System startup with a constant resistive load. Simulated (**a**,**c**) and experimental (**b**,**d**). Two operation modes in steady-state are shown: (**a**,**b**) boost mode ($V_{g}=$ 200 V, $v_{o}=$ 293 V and and $R_{o}=$ 200 Ω) and (**c**,**d**) buck mode ($V_{g}=$ 350 V, $v_{o}=$ 293 V and $R_{o}=$ 32.3 Ω). CH1 or CH2: $v_{o}$ (100 V/div), CH3: $i_{g}$ (2 A/div), CH4: $i_{L}$ ( 2A/div), and time base of 4ms.

**Figure 8 sensors-21-05712-f008:**
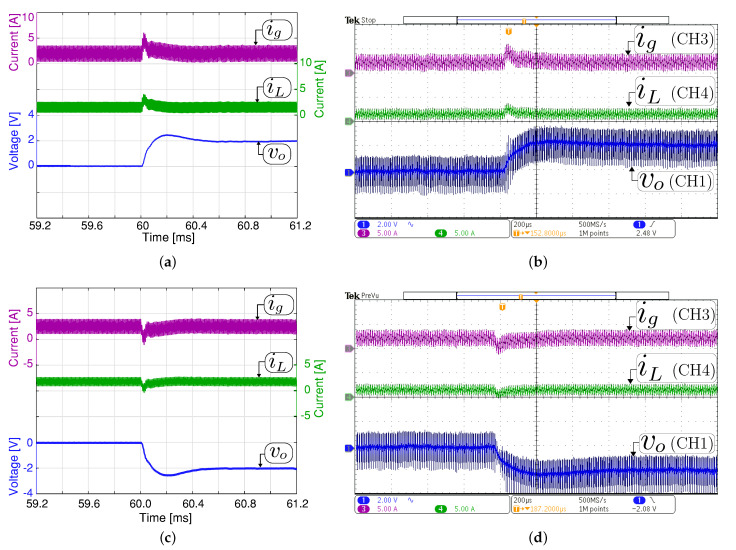
Small signal transient response with a constant resistive load $R_{o}=$ 200 Ω in boost mode ($V_{g}=$ 200 V). Simulated (**a**,**c**) and experimental (**b**,**d**). Transient response when the output voltage reference changes from 294 to 296 V (**a**,**b**), and from 296 V to 294 V (**c**,**d**). CH1: $v_{o}$ (2 Vac/div), CH3: $i_{g}$ (5 A/div), CH4: $i_{L}$ (5 A/div), and time base of 200 μs.

**Figure 9 sensors-21-05712-f009:**
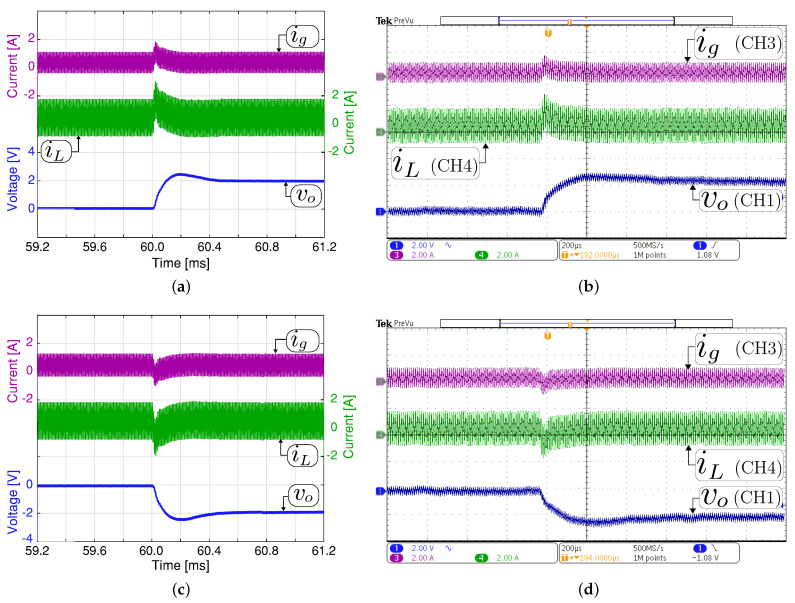
Small signal transient response with a constant resistive load $R_{o}=$ 200 Ω in buck mode ($V_{g}=$ 200 V). Simulated (**a**,**c**) and experimental (**b**,**d**). Transient response when the output voltage reference changes from 98 to 100 V (**a**,**b**), and from 100 V to 98 V (**c**,**d**). CH1: $v_{o}$ (2 Vac/div), CH3: $i_{g}$ (2 A/div), CH4: $i_{L}$ (2 A/div), and time base of 200 μs.

**Figure 10 sensors-21-05712-f010:**
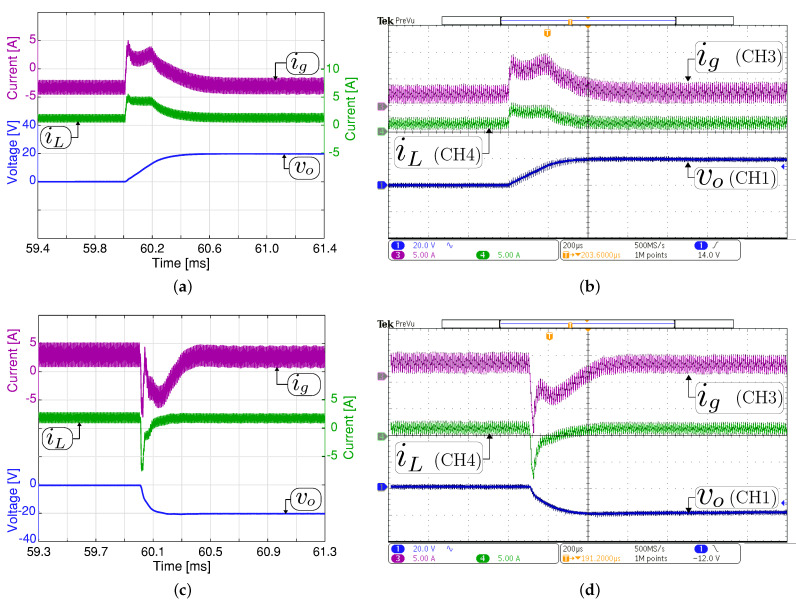
Large signal transient response with a constant resistive load $R_{o}=$ 200 Ω in boost mode ($V_{g}=$ 200 V). Simulated (**a**,**c**) and experimental (**b**,**d**). Transient response when the output voltage reference changes from 294 to 314 V (**a**,**b**), and from 314 to 294 V (**c**,**d**). CH1: $v_{o}$ (20 Vac/div), CH3: $i_{g}$ (5 A/div), CH4: $i_{L}$ (5 A/div), and time base of 200 μs.

**Figure 11 sensors-21-05712-f011:**
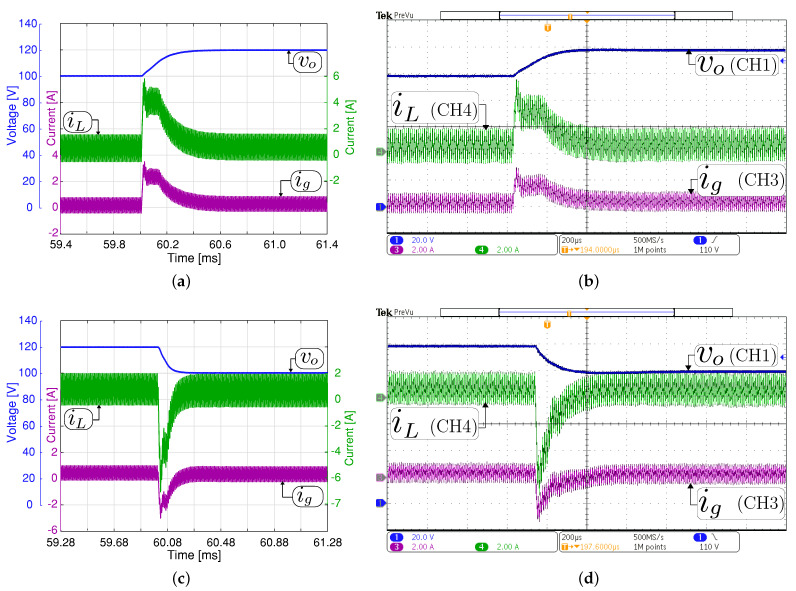
Large signal transient response with a constant resistive load $R_{o}=$ 200 Ω in buck mode ($V_{g}=$ 200 V). Simulated (**a**,**c**) and experimental (**b**,**d**). Transient response when the output voltage reference changes from 100 to 120 V (**a**,**b**), and from 120 to 100 V (**c**,**d**). CH1: $v_{o}$ (20 V/div), CH3: $i_{g}$ (2 A/div), CH4: $i_{L}$ (2 A/div), and time base of 200 μs.

**Figure 12 sensors-21-05712-f012:**
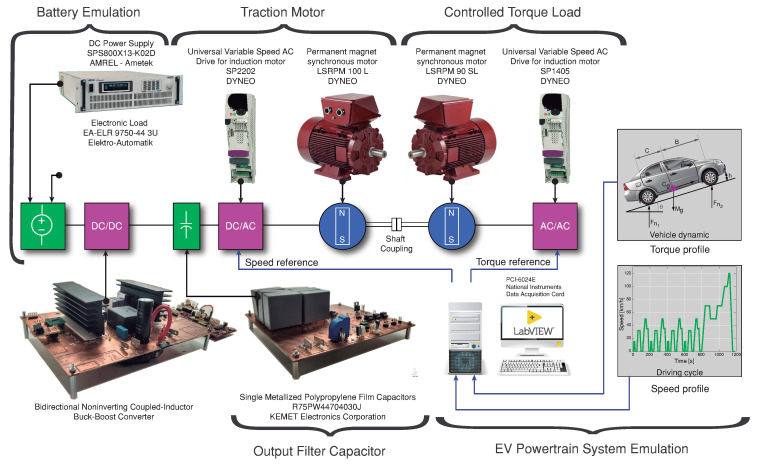
Diagram of the experimental setup: Converter dc-dc and EV powertrain.

**Figure 13 sensors-21-05712-f013:**
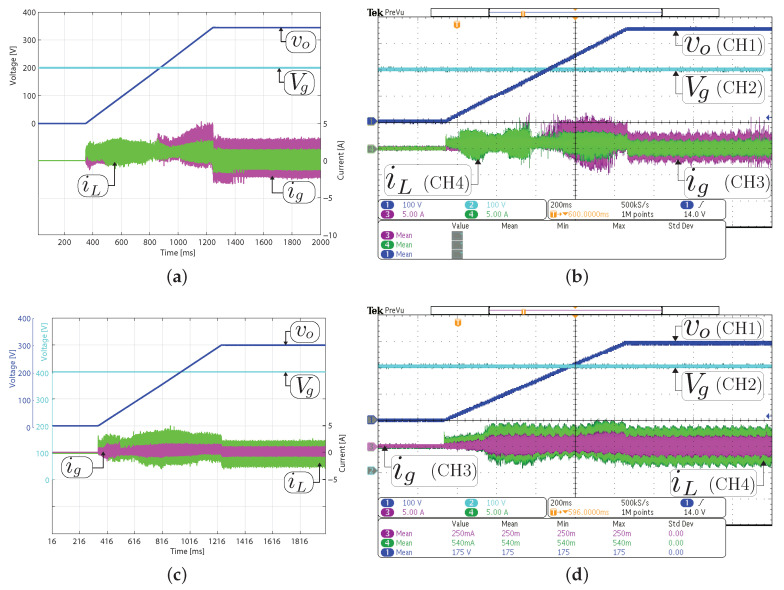
System startup with an EV powertrain system emulation. Simulated (**a**,**c**) and experimental (**b**,**d**). Two operation modes in steady-state are shown: (**a**,**b**) boost mode ($V_{g}=$ 200 V, $v_{o}=$ 350 V) and (**c**,**d**) buck mode ($V_{g}=$ 400 V, $v_{o}=$ 300 V). CH1: $v_{o}$ (100 V/div), CH2: $V_{g}$ (100 V/div), CH3: $i_{g}$ (5 A/div), CH4: $i_{L}$ (5 A/div), and time base of 200ms.

**Figure 14 sensors-21-05712-f014:**
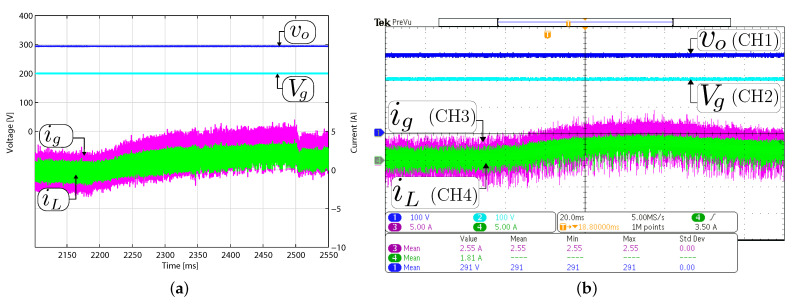
Boost operation mode in steady-state for a step power transition ($P_{o}=$ 300 to 750 W, $V_{g}=$ 200 V, $\;V_{o}=$ 300 V) with an EV powertrain system emulation. Simulated (**a**), and experimental (**b**). CH1: $V_{o}$ (100 V/div), CH2: $V_{g}$ (100 V/div), CH3: $i_{g}$ (5 A/div), CH4: $i_{L}$ (5 A/div), and time base of 20ms.

**Figure 15 sensors-21-05712-f015:**
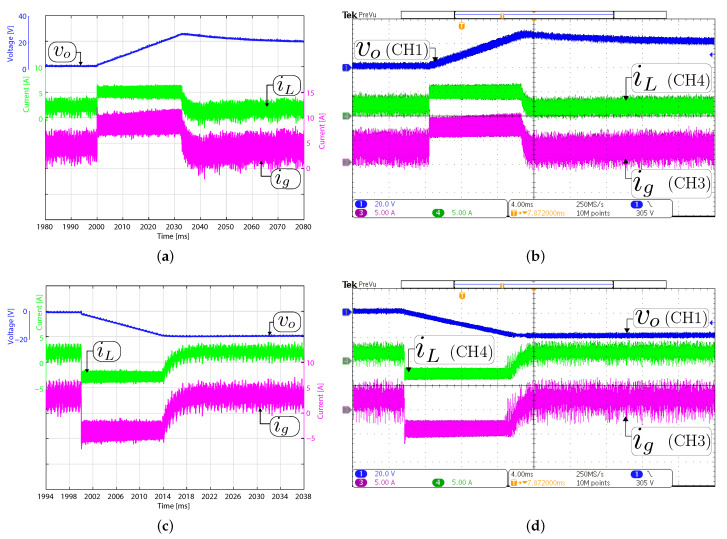
Boost operation mode in steady-state with an EV powertrain system emulation $V_{g}=$ 200 V. Transient response when the output voltage reference changes from 300 to 320 V. Simulated (**a**,**c**) and experimental (**b**,**d**), time base of 10ms (**a**,**b**), and from 320 to 300 V, time base of 4ms (**c**,**d**). CH1: $V_{o}$ (20 V/div), CH3: $i_{g}$ (5 A/div), CH4: $i_{L}$ (5 A/div).

**Table 1 sensors-21-05712-t001:** Slope of the output current waveform.

Mode	$m_{1}$	$-m_{2}$
Buck	$\frac{M\left(V_{g}-v_{c}\right)-L\left(v_{o}-v_{c}\right)}{L^{2}-M^{2}}$	$\frac{M\left(V_{g}-v_{c}\right)-Lv_{o}}{L^{2}-M^{2}}$
Boost	$\frac{MV_{g}-L\left(v_{o}-v_{c}\right)}{L^{2}-M^{2}}$	$\frac{M\left(V_{g}-v_{c}\right)-L\left(v_{o}-v_{c}\right)}{L^{2}-M^{2}}$

**Table 2 sensors-21-05712-t002:** CF and PM of voltage loop gain.

Mode	Simulated		Experimental	
	CF	PM	CF	PM
	[kHz]	[deg]	[kHz]	[deg]
Boost	1.99	63.79	2.03	79.4
Buck	1.99	66.52	1.94	71.2

**Table 3 sensors-21-05712-t003:** Selected components and parameters for the buck–boost converter.

Parameter	Value or Type
Input voltage $V_{g}$	200–400 V
Output voltage $V_{o}$	100–400 V
Rated Power	1.6 kW
Switching frequency $f_{s}$	100 kHz
Output capacitor $C_{o}$	6× R75PW44704030J, 28 μF, 630 V
Damping capacitor $C_{d}$	MKP1848S62070JP2F, 20 μF, 700 V
Intermediate capacitor *C*	4× R76PN33304030J, 1.32 μF, 630 V
Coupled inductor	M=135 μH and L=270 μH,
	Core: 77,908 Magnetics,
	Number turns: 80,
	Wire size: 18 AWG.
Damping resistance $R_{d}$	2× BPR10100J in parallel, 5 Ω,
	10 W, 500 V
MOSFET Driver	UCC27714D
Power semiconductors $Q_{1}-Q_{4}$	SCT2450KEC

**Table 4 sensors-21-05712-t004:** Parameters of the PMSMs.

Parameter	90 SL	100 L
Motor rated speed	1500 [rpm]	1500 [rpm]
Number of pole pairs	8	8
Stator resistance $R_{s}$	2.34 Ω	1.277 Ω
*d*-axis inductance $L_{d}$	50.124 mH	29.128 mH
*q*-axis inductance $L_{d}$	29.128 mH	19.295 mH
Moment of inertia *J*	0.0032 kg·m${}^{2}$	0.0066 kg·m${}^{2}$
Electrical constant $k_{e}$	212 $V_{kp}/k_{rpm}$	223 $V_{kp}/k_{rpm}$

## Data Availability

Not applicable.

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
