# Peer review of "A Bidirectional Versatile Buck–Boost Converter Driver for Electric Vehicle Applications"

_sensors, 2021, doi:10.3390/s21175712_

Round 1
Reviewer 1 Report
Dear authors,
The reviewer has no further comments, the authors have satisfactorily explined my comments.
Author Response
Author response: Thank you very much for your explanation. We are satisfied to have clarified your comments.

Reviewer 2 Report
It is hard to say that You have a new converter when You are using H-bridge topology. This approach reminds me to:
https://ieeexplore.ieee.org/stamp/stamp.jsp?tp=&arnumber=4455453
Traditional Buck and Boost converters were replaced with H-bridge topology in power MOSFET driver. It seems that is hard to be absolute original.
Well done.
Author Response
Author response: We thank the reviewer for the opportunity to clarify this point. The versatile buck-boost converter is a new topology proposed in [26] with different topology modifications throughout all the investigations carried out to date, as shown in Fig. 4:
[26] Restrepo, C.; Calvente, J.; Cid-Pastor, A.; El Aroudi, A.; Giral, R. A noninverting buck–boost DC–DC switching converter with high efficiency and wide bandwidth. IEEE Transactions on Power Electronics 2011, 26, 2490–2503.
[27] Restrepo, C.; Calvente, J.; Romero, A.; Vidal-Idiarte, E.; Giral, R. Current-Mode Control of a Coupled-Inductor Buck–Boost DC–DC Switching Converter. IEEE Transactions on Power Electronics 2012, 27, 2536–2549. doi:10.1109/TPEL.2011.2172226.
[28] Restrepo, C.; Konjedic, T.; Calvente, J.; Milanovic, M.; Giral, R. Fast Transitions Between Current Control Loops of the Coupled-Inductor Buck–Boost DC–DC Switching Converter. IEEE Transactions on Power Electronics 2013, 28, 3648–3652. doi:10.1109/TPEL.2012.2231882.
[29] Restrepo, C.; Konjedic, T.; Calvente, J.; Giral, R. Hysteretic transition method for avoiding the dead-zone effect and subharmonics in a noninverting buck–boost converter. IEEE Transactions on Power Electronics 2015, 30, 3418–3430.
[30] Ramírez-Murillo, H.; Restrepo, C.; Konjedic, T.; Calvente, J.; Romero, A.; Baier, C.R.; Giral, R. An Efficiency Comparison of Fuel-Cell Hybrid Systems Based on the Versatile Buck–Boost converter. IEEE Transactions on Power Electronics 2018, 33, 1237–1246. doi: 10.1109/TPEL.2017.2678160.
[31] Ramírez-Murillo, H.; Restrepo, C.; Calvente, J.; Romero, A.; Giral, R. Energy Management of a Fuel-Cell Serial–Parallel Hybrid System. IEEE Transactions on Industrial Electronics 2015, 62, 5227–5235.
[32] Ramírez-Murillo, H.; Restrepo, C.; Calvente, J.; Romero, A.; Giral, R. Energy Management DC System Based on Current-Controlled Buck-Boost Modules. IEEE Transactions on Smart Grid 2014, 5, 2644–2653.
[33] Restrepo, C.; Konjedic, T.; Flores-Bahamonde, F.; Vidal-Idiarte, E.; Calvente, J.; Giral, R. Multisampled Digital Average Current Controls of the Versatile Buck-Boost Converter. IEEE Journal of Emerging and Selected Topics in Power Electronics 2018.
[34] Restrepo, C.; Garcia, G.; Flores-Bahamonde, F.; Murillo-Yarce, D.; Guzman, J.I.; Rivera, M. Current Control of the Coupled-Inductor Buck–Boost DC–DC Switching Converter using a Model Predictive Control Approach. IEEE Journal of Emerging and Selected Topics in Power Electronics 2020, 3348- 3360.
[35] Méndez-Díaz, F.; Pico, B.; Vidal-Idiarte, E.; Calvente, J.; Giral, R. HM/PWMSeamless Control of a Bidirectional Buck–Boost Converter for a Photovoltaic Application. IEEE Transactions on Power Electronics 2019, 34, 2887–2899.
The authors believe that there may be confusion with the word "driver" since it can be used interchangeably to refer to a circuit or component used to control another circuit or component. However, there are two terms when the driver word is commonly used:
- MOSFET Gate Driver
It is a specialized circuit used to drive the gate (gate driver) of power MOSFETs effectively and efficiently in high-speed switching applications. A conventional gate drive circuit to drive a power metal-oxide semiconductor field-effect transistor (MOSFET) is shown in the following figure [Ref 1]:
[Ref 1] W. Eberle, Z. Zhang, Y. Liu and P. C. Sen, "A Current Source Gate Driver Achieving Switching Loss Savings and Gate Energy Recovery at 1-MHz," in IEEE Transactions on Power Electronics, vol. 23, no. 2, pp. 678-691, March 2008, doi: 10.1109/TPEL.2007.915769.
- Motor driver
An electric motor is a device that converts electrical energy to mechanical energy. It also can be viewed as a device that transfers energy from an electrical source to a mechanical load. The system in which the motor is located and makes it spin is called the drive also referred to as the electric drive or motor drive. The motor drive function is to draw electrical energy from the electrical source and supply electrical energy to the motor, so that the desired mechanical output is achieved [Ref 2]. Typically, this is the speed of the engine, torque, and position of the motor shaft. The following figure shows the block diagram of a motor drive:
[Ref 2] Power Electronics in Motor Drives: Where is it?, Texas Instruments Report, October 2019.
Therefore, the article that the reviewer mentions ([Ref 1]) refers to a new gate drive circuit for power MOSFETs as is shown in the following figure:
However, the article is focused on a motor driver as shown in Figure 12 of the article:

Reviewer 3 Report
The manuscript presents an improvement of the buck-boost converter for application in electrical vehicles. The manuscript is well structured and has a simulation and experimental results. The novelty of the paper is focused on improvements applied for the specific application of the named converter.
I have doubts regarding how the topic of the manuscript fits the MDPI Sensors scope.
Please find comments regarding the manuscript:
Figure 1: "power converter" term is used, but in-text "DC-AC traction inverter" is used
line 36: dc-dc
line 53: abbriviation ZVS not defined
line 82: abbreviation EV (electric vehicle) was defined, but not in use
lines 94-96: authors write "/…/The whole system is tested experimentally in a 1.6 kW prototype /…/" What are the other parameters of the prototype, e.g., voltage levels
Figure 4: seems to be very interesting but not adequately analyzed in the text
line 164: Table 3 mentioned in the text before Table 2
Figure 6: it would be more reasonable to have simulated and experimental Bode plots on the same diagram for different modes
Figures 7-11: traces axis are too small, and it is hard to understand
Lines 228-232: Please explain more in detail the power distribution in the experimental testbench. It is confusing that studied buck-boost is 1.6 kW, traction motor 4.5 kW, and loading motor is 3 kW. In my opinion, it is impossible to reach high efficient system with such power distribution. Moreover, the motor under test is underloaded, it also can be seen from the further description of the experimental part.
Moreover, I am confused about how the term "high voltage" is used in the manuscript. According to IEC standards, for voltage level over 1500VDC is defined as high voltage. However, the target voltage level is 400VDC what is the range of the main battery in an electric vehicle. The main battery sometimes called a high-voltage battery because there is an additional auxiliary battery, usually 12VDC. Therefore, please use correct terms in the manuscript or define the "high voltage" according to how you use it in the text.
Author Response
Reviewer#3, Concern # 1: The manuscript presents an improvement of the buck-boost converter for application in electrical vehicles. The manuscript is well structured and has a simulation and experimental results. The novelty of the paper is focused on improvements applied for the specific application of the named converter.
I have doubts regarding how the topic of the manuscript fits the MDPI Sensors scope.
Author response: Thanks for raising this concern. We send the manuscript to the Special Issue titled “Advances in Intelligent Vehicle Control” of Sensors. You can see the special issue description in the following link:
https://www.mdpi.com/journal/sensors/special_issues/intelligent_vehicle_sensors
Among the topics declared by the Special Issue Editor are included "Driver-Vehicle Systems" and "Energy Management Strategies for Hybrid and Electric Vehicles". Our research reported in the article covers both topics, which is why we consider that our work is in the scope of the Sensors journal.
Reviewer#3, Concern # 2: Figure 1: "power converter" term is used, but in-text "DC-AC traction inverter" is used. line 36: dc-dc
Author response: Following the reviewer’s observation, we added a description (lines 37, 38 and 39) to clarify this term.
Reviewer#3, Concern # 3: line 53: abbriviation ZVS not defined
Author response: Thank you for noticing, we added the corresponding abbreviation.
Reviewer#3, Concern # 4: line 82: abbreviation EV (electric vehicle) was defined, but not in use
Author response: We do not understand the reviewer's comment. Electric vehicles (EVs) were used seven times in the whole article, while the acronym of Electric vehicle (EV) was employed thirty-three times in the manuscript.
Reviewer#3, Concern # 5: lines 94-96: authors write "/…/The whole system is tested experimentally in a 1.6 kW prototype /…/" What are the other parameters of the prototype, e.g., voltage levels
Author response: Please note that the most detailed information about the prototype is described in the Simulation and experimental results section. In addition, the selected components and parameters for the power converter are summarized in Table 3.
Reviewer#3, Concern # 6: Figure 4: seems to be very interesting but not adequately analyzed in the text
Author response: The comment is very appropriate. The new article is linking the main contributions (lines 109 to 125) of the work with Figure 4.
Reviewer#3, Concern # 7: line 164: Table 3 mentioned in the text before Table 2
Author response: Thank you for noticing. We changed the text in the new version of the paper and Table 3 is referenced only in experimental results section.
Reviewer#3, Concern # 8 Figure 6: it would be more reasonable to have simulated and experimental Bode plots on the same diagram for different mode
Author response: Following the reviewer’s observation, we changed Figure 6.
Reviewer#3, Concern # 9 Figures 7-11: traces axis are too small, and it is hard to understand
Author response: Thank you for noticing. Updated versions of Figs. 7 to 11 are now provided where several changes were made to legends, ticks, and labels of the simulation results to facilitate their understanding. The aspect ratio of these vectorial graphs has also been normalized.
Reviewer#3, Concern # 10 Lines 228-232: Please explain more in detail the power distribution in the experimental testbench. It is confusing that studied buck-boost is 1.6 kW, traction motor 4.5 kW, and loading motor is 3 kW. In my opinion, it is impossible to reach high efficient system with such power distribution. Moreover, the motor under test is underloaded, it also can be seen from the further description of the experimental part}
Author response: The reviewer is absolutely right. A description about the reviewer’s observation has been added in subsection 4.4 in lines from 239 to 244. In addition, in the conclusions, the converter parallelization is proposed as future work.
Reviewer#3, Concern # 11: Moreover, I am confused about how the term "high voltage" is used in the manuscript. According to IEC standards, for voltage level over 1500VDC is defined as high voltage. However, the target voltage level is 400VDC what is the range of the main battery in an electric vehicle. The main battery sometimes called a high-voltage battery because there is an additional auxiliary battery, usually 12VDC. Therefore, please use correct terms in the manuscript or define the "high voltage" according to how you use it in the text.
Author response:
Thank you very much for your concern. In a general way, the International Electrotechnical Commission and its national counterparts define high voltage as above 1500 V for direct current. However, the numerical definition of high voltage depends on the context. In the case of the automotive industry, high-voltage refers to voltages above 60 V. For more information, please consult the voltage classes for electric mobility report published by ZVEI - German Electrical and Electronic Manufacturers’ Association Centre of Excellence Electric Mobility:
https://www.zvei.org/fileadmin/user_upload/Presse_und_Medien/Publikationen/2014/april/Voltage_Classes_for_Electric_Mobility/Voltage_Classes_for_Electric_Mobility.pdf

Reviewer 4 Report
- The justification of why step up converters are utilized in EVs provided by the authors in Section 1 does not seem sound. The authors seem to imply that step up converters are used to mitigate the risks of battery failure – however reference [4] which the authors use to justify this is about using an MMC converter (with battery cells as the submodule energy sources) as a traction drive. In [4] the cell balancing functions normally performed by a BMS were conducted via control of the MMC. I recommend that the authors provide better justification for implementing step-up converters between the battery and traction inverter.
- On page 4, I don’t think that validation via simulation and experiments can be counted as one of the novel contributions of a manuscript.
- The authors should include a section describing the design methodology of the coupled inductor used in their converter.
- The authors should elaborate more on the impact of the parasitic capacitance reduction in this design.
- The manuscript would by providing a comparison of the performance achieved with this design with the previous iterations of this topology.
Author Response
Reviewer#4, Concern # 1 The justification of why step up converters are utilized in EVs provided by the authors in Section 1 does not seem sound. The authors seem to imply that step up converters are used to mitigate the risks of battery failure – however reference [4] which the authors use to justify this is about using an MMC converter (with battery cells as the submodule energy sources) as a traction drive. In [4] the cell balancing functions normally performed by a BMS were conducted via control of the MMC. I recommend that the authors provide better justification for implementing step-up converters between the battery and traction inverter.
Author response: Thank you very much for your observation. We added in the introduction more arguments that justify the importance of using a dc-dc converter after the battery system to improve the performance of the automotive traction systems
Reviewer#4, Concern # 2: On page 4, I don’t think that validation via simulation and experiments can be counted as one of the novel contributions of a manuscript.
Author response: Thank you very much for your concern. We eliminated this item of the contribution part.
Reviewer#4, Concern # 3: The authors should include a section describing the design methodology of the coupled inductor used in their converter.
Author response: Detailed information about the proposed convert's design and efficiency analysis is widely studied in [41]. Therefore, we added a description in the simulation and experimental results section to refer the reader to this paper for more information about the coupled inductor design and efficiency results.
[41] González-Castaño, C.; Restrepo, C.; Giral, R.; García-Amoros, J.; Vidal-Idiarte, E.; Calvente, J. Coupled inductors design of the bidirectional non-inverting buck–boost converter for high-voltage applications. IET Power Electronics 2020.
Reviewer#4, Concern # 4: The authors should elaborate more on the impact of the parasitic capacitance reduction in this design.
Author response: The impact of the parasitic capacitance reduction in the proposed power converter is studied deeply in the previously published work [41]. The focus of the new article is to demonstrate the converter performance in the EV powertrain system application.
[41] González-Castaño, C.; Restrepo, C.; Giral, R.; García-Amoros, J.; Vidal-Idiarte, E.; Calvente, J. Coupled inductors design of the bidirectional non-inverting buck–boost converter for high-voltage applications. IET Power Electronics 2020.
Reviewer#4, Concern # 5: The manuscript would by providing a comparison of the performance achieved with this design with the previous iterations of this topology.
Author response: As previously mentioned, the focus of the new article is to demonstrate the outstanding converter performance in an EV powertrain system application. The comparison between both topologies proposed by the reviewer has been widely analyzed with experimental and simulation results in [41].
[41] González-Castaño, C.; Restrepo, C.; Giral, R.; García-Amoros, J.; Vidal-Idiarte, E.; Calvente, J. Coupled inductors design of the bidirectional non-inverting buck–boost converter for high-voltage applications. IET Power Electronics 2020.

Round 2
Reviewer 4 Report
The authors have stated in the reviewer response that the purpose of this article is "To demonstrate the outstanding converter performance in an EV powertrain system application" of the topology in question.
In this reviewer's opinion, the introduction of an existing topology to a new application area is of insufficient novelty for a research paper.